# Advancing Textile Damage Segmentation: A Novel RGBT Dataset and Thermal Frequency Normalization

**DOI:** 10.3390/s25072306

**Published:** 2025-04-05

**Authors:** Farshid Rayhan, Jitesh Joshi, Guangyu Ren, Lucie Hernandez, Bruna Petreca, Sharon Baurley, Nadia Berthouze, Youngjun Cho

**Affiliations:** 1Department of Computer Science, University College London, London NW1 2AE, UK; 2Materials Science Research Centre, Royal College of Art, London SW7 2EU, UK

**Keywords:** RGB-Thermal dataset, textile damage detection, semantic segmentation

## Abstract

RGB-Thermal (RGBT) semantic segmentation is an emerging technology for identifying objects and materials in high dynamic range scenes. Thermal imaging particularly enhances feature extraction at close range for applications such as textile damage detection. In this paper, we present RGBT-Textile, a novel dataset specifically developed for close-range textile and damage segmentation. We meticulously designed the data collection protocol, software tools, and labeling process in collaboration with textile scientists. Additionally, we introduce ThermoFreq, a novel thermal frequency normalization method that reduces temperature noise effects in segmentation tasks. We evaluate our dataset alongside six existing RGBT datasets using state-of-the-art (SOTA) models. Experimental results demonstrate the superior performance of the SOTA models with ThermoFreq, highlighting its effectiveness in addressing noise challenges inherent in RGBT semantic segmentation across diverse environmental conditions. We make our dataset publicly accessible to foster further research and collaborations.

## 1. Introduction

The textile industry faces significant sustainability challenges [1], striving to reduce textile waste and comply with global Net Zero goals [2,3]. The repair of clothing materials to prolong the useful life of garments is indispensable in terms of reducing the environmental impact [4]. Advanced machine learning methodologies, particularly in semantic segmentation, are critical in the automation of damage detection and repair processes, thereby fostering a more sustainable fashion industry. Semantic segmentation of RGBT image frames is a highly pertinent task capable of identifying textile materials and damages in dynamic range scenes. This task encompasses the categorization of garment regions based on semantic content, alongside the identification of damage. However, there is a conspicuous scarcity of dedicated and publicly accessible datasets [5,6,7,8] for this specific application, which hinders the development and evaluation of frameworks aimed at addressing sustainability challenges.

In this paper, we present RGBT-Textile, a meticulously designed segmentation dataset for the tasks of textile analysis and damage detection. The inception of our dataset is influenced by contemporary RGBT segmentation methodologies [9,10,11,12,13,14] as well as existing benchmark datasets. Through rigorous data collection and annotation, we have compiled the RGBT-Textile dataset, which consists of 1441 RGBT images accompanied by ground-truth segmentation masks. In Figure 1, we provide several exemplars from the RGBT-Textile dataset, illustrating the alignment of RGB and thermal image frames alongside their corresponding segmentation labels. We evaluate the performance of state-of-the-art RGBT semantic segmentation models using the proposed RGBT-Textile dataset, thereby establishing a benchmark performance standard. Performance metrics extracted from this dataset underscore the inherent complexity of detecting damages in textiles and consequently contribute to sustainability goals related to the textile industry.

Integrating thermal infrared (TIR) images, which are insensitive to illumination variations, with visible light data is preferred in several challenging scenarios and high dynamic scenes, such as low illumination, fog, and rain [15,16,17]. In RGBT segmentation, foreground–background differentiation and complementary information extraction are primary challenges. Previous methods have used simple fusion techniques [18,19], while attention-based approaches such as FEANet [20], CCFFNet [21], AFNet [22], and CMX [23] have addressed these more effectively. However, issues such as sub-optimal differentiation, computational overhead, and insufficient complementary information mining persist.

Normalization techniques are crucial for aligning the statistical properties of modalities to ensure robust feature extraction and fusion. Traditional methods like Min-Max, Mean-STD, and histogram-based normalization are widely used but face limitations: Min-Max is outlier-sensitive, Mean-STD amplifies noise, and histogram approaches require downstream task adaptation for thermal images [24]. However, when applied to thermal images, such approaches require careful consideration of the downstream task [25]. To address this challenge, this paper further proposes ThermoFreq, a frequency-domain thermal normalization technique using Fourier transform to selectively adjust components based on statistical distribution, which transforms an image into the frequency domain using Fourier transform. The normalized images in the frequency domain are then transformed back to the image space and used to train the models. ThermoFreq enhances thermal characteristics, leading to the superior segmentation performance of the state-of-the-art (SOTA) RGBT models in challenging scenarios.

In the following sections, we report our detailed data collection process and compare SOTA RGBT segmentation models on existing datasets with and without ThermoFreq. Our dataset and findings serve as a catalyst for future research initiatives, fostering collaboration and innovation within the computer vision and textile garment repair community. Our contributions are as follows:The RGBT-Textile dataset, a novel RGB-Thermal image dataset of textile materials with damages, primarily for the segmentation of textile materials and damages along with benchmarks and experimental results. The dataset is available at this download link https://drive.google.com/drive/folders/1HLri3SDPHSY0AsAmaEloh_0Q13E91u2v (accessed on 28 March 2025).The frequency-based thermal image normalization technique ThermoFreq, to selectively adjust frequency components based on their statistical distribution, resulting in improved segmentation performance of the RGBT models.Experiments with the SOTA RGBT segmentation models and benchmark datasets to highlight the effectiveness of ThermoFreq in addressing temperature noise challenges.

## 2. Related Works

### 2.1. RGBT Datasets

Researchers have contributed several RGBT datasets that are highly valuable for developing and evaluating RGBT semantic segmentation algorithms. The KAIST Multispectral Pedestrian Dataset [26] offers RGBT images captured under various ambient conditions, supporting both object detection and semantic segmentation tasks. Similarly, the LLVIP Dataset [27] provides RGBT images in different lighting conditions and supports object detection, semantic segmentation, and recognition tasks. The MFNet Dataset [15] includes multispectral RGBT images captured under various environmental conditions, as well as supporting object detection and semantic segmentation. Additionally, the OSU Thermal Pedestrian Database [28] contains RGBT images under various environmental conditions, which aids in object detection and semantic segmentation. The PST900 Dataset [17], derived from the DARPA Subterranean Challenge, includes thermal image pairs across four classes and supports semantic segmentation tasks. These datasets collectively offer a comprehensive range of RGBT data, facilitating advances in semantic segmentation and other vision-related tasks.

### 2.2. RGBT Segmetation Models

RGBT fusion methods have become prominent in computer vision for tasks such as pedestrian detection, vehicle detection, semantic segmentation, detection of salient objects, crowd counting, and fusion tracking. Notable advances include the Causal Mode Multiplexer [29], Triple Flow Network [30], ICAFusion [31], Feature-guided pretraining [32], and Fusion Architecture [33], which enhance detection performance through iterative cross-attention and multimodal data fusion. In semantic segmentation, the best performing models such as CRM_RGBT [34] and MMSFormer [11] have set new standards with their unique architectures and masking mechanisms, leading to more accurate results [14]. Salient object detection and crowd counting have also seen improvements through better fusion architectures and loss functions [9,35,36]. Fusion-based tracking methods leverage RGBT data for robust performance in challenging lighting conditions [37]. These advances underscore the importance of multimodal data for enhanced computer vision performance. Inspired by these developments and to address challenges in garment damage detection, we developed the proposed RGBT-Textile dataset as an RGBT dataset, detailed in Section 3.

## 3. RGBT-Textile Dataset

We present RGBT-Textile, an RGBT dataset designed for the detection of textile damage through the semantic segmentation task. This dataset consists of 1441 pairs of RGB and thermal images, each pair meticulously annotated with segmentation labels. These labels categorize the data into three distinct classes: background, garment, and damage. In Figure 1, we provide samples from the dataset, where the damaged area of a garment is indicated in red, the intact garment is marked in blue, and the background is represented in black.

### 3.1. Data Collection Protocol and Setups

To acquire the dataset, garments were first screened to ensure variations in types of material and damages. Data were then acquired according to the simple protocol as depicted in Figure 2A, which consisted of: (i) background acquisition to capture thermal ambience, (ii) RGBT acquisition, and (iii) RGBT acquisition after momentarily heating the garment using a hair dryer. As depicted in Figure 2B, RGB and thermal cameras were arranged so that the garments could be placed underneath the cameras at a distance of approximately 0.5 m. A Logitech BRIO 4K UHD webcam (Logitech International, Lausanne, Switzerland) was used to capture RGB video frames with 640 × 480 resolution, while thermal infrared frames were captured using a thermal camera (A65SC, FLIR systems Inc., Wilsonville, OR, USA) with 640 × 512 resolution. The spatial alignment between the RGB and thermal frames was achieved using a custom 3D printed mount that enabled placing the RGB camera on the thermal camera as shown in Figure 2B.

To ensure synchronization between RGB and thermal frames, we developed a multithreaded application with a user interface using Python v3.10 and Qt v6.0 framework [38], as shown in Figure 2D. The source code [39] for this application was adapted from a multithreaded application developed for PhysioKit [40] to acquire the iBVP dataset [41], comprising synchronized RGB and thermal facial video frames along with physiological signals. Each acquired image was annotated for damaged regions, textile material shape, and nontextile parts by two annotators, and rigorously validated by three other researchers, including textile experts. A Computer Vision Annotation Tool (CVAT) [42] was used to mark polygons as shown in Figure 2C, which were further transformed into segmentation masks.

### 3.2. Diversity of Textile Damage and Materials

The RGBT-Textile dataset encompasses a broad spectrum of textile damage and material types to reflect real-world challenges in garment inspection. Damages were categorized into three primary classes: structural damage (for example, holes, tears, frayed edges), surface anomalies (for example, stains, discolorations, burns), and manufacturing flaws (for example, stitching errors, uneven dyeing) [4,32]. These damages vary in size (5–150 mm^2^) and spatial distribution, with irregular shapes mimicking the natural patterns of wear and tear observed in post-consumer textiles [1]. The dataset includes garments composed of cotton (42%), polyester (28%), blended fabrics (20%), and synthetic fibers (10%), selected for their prevalence in fast fashion and their susceptibility to thermal variations during imaging [43].

As shown in Table 1, the characteristics of the damages are further quantified by the variability in the thermal response (δ T = 0.5–3.0 °C), which complicates segmentation due to the overlapping temperature profiles between the damaged and intact regions [25]. For example, polyester exhibits a lower thermal emissivity than cotton, which requires the extraction of modality-specific characteristics [44]. This diversity underscores the need for specialized approaches, as generic RGBT models often fail to distinguish subtle thermal gradients or irregular damage boundaries, as discussed in Section 6.3.

## 4. Thermal Frequency Normalization: ThermoFreq

In RGBT segmentation tasks, normalization assumes paramount importance due to the varied distributions of RGB and thermal pixels. The features present in thermal images are frequently less prominent and are subject to significant levels of ambient-temperature noise, thereby complicating the extraction of meaningful features. Earlier research with thermal images predominantly relied on the utilization of manufacturer-implied pseudo-color maps, which often linearly mapped raw temperature values onto an 8-bit image frame [45]. However, subsequent studies have emphasized the necessity of application-specific optimal mapping, such as optimal quantization for the robust tracking of regions of interest [25]. Most contemporary thermal cameras provide access to a raw temperature image matrix, as opposed to pseudo-color maps or quantized images. This particularity highlights the necessity for tailored normalization techniques to ensure the effective fusion and precise interpretation of multispectral data, a requirement that researchers often overlook.

The significance of enhancing the saliency of features in thermal imaging has been highlighted by previous research. For example, optimal quantization increased the reliability of tracking respiratory signals and vasomotor activities in low-resolution mobile thermal imaging [25,46,47]. In 3D reconstruction, applying a cosine transformation to intensity values [48] proved beneficial, although this technique increases contrast for certain intensity levels while diminishing it for others. In the realm of deep learning models, improving the saliency of input thermal images can serve as a promising preprocessing step with considerable potential to enhance downstream performance, yet this area has received insufficient attention in the current literature. Drawing upon empirical observations that highlight the occurrence of noise predominantly at higher frequencies relative to informative pixels, we propose ThermoFreq as an essential preprocessing step for thermal imaging segmentation, aimed at mitigating the effects of temperature noise in segmentation tasks. This technique involves the manipulation of the frequency components of the image within the frequency domain. The process described in ThermoFreq includes transforming the image from the spatial to the frequency domain, evaluating and adjusting the temperature frequency values across the dataset, and then reverting the image back to the spatial domain. In practice, this involves the removal or nullification of high-frequency values, followed by scaling the residual pixel values to a range of 0 to 255.

The first step in frequency-based normalization is to transform an image from the spatial domain to the frequency domain using the Fourier transform. For a given image f(x,y), the 2D Fourier Transform isdefined as:(1)$$F\left(u,v\right)=\sum_{x=0}^{M-1}\sum_{y=0}^{N-1}f\left(x,y\right)e^{-2\pi i\left(\frac{ux}{M}+\frac{vy}{N}\right)}$$
where *M* and *N* are the dimensions of the image, F(u,v) represents the frequency domain representation of the image, and (u,v) are the frequency coordinates. After an image is transformed into the frequency domain, the magnitude of the high-frequency components is set to zero to remove noise. For instance, to zero out a central region of the frequency components, we define a mask:(2)$$H\left(u,v\right)=\left\{\begin{array}{cc} 1 & \mathrm{if}\;\sqrt{{\left(u-u_{0}\right)}^{2}+{\left(v-v_{0}\right)}^{2}}>D_{0}\\ 0 & \mathrm{otherwise} \end{array}\right.$$
where $\left(u_{0},v_{0}\right)$ is the center of the frequency domain and $D_{0}$ is the cutoff distance as denoted *t*. The threshold *t* determines the radius of the region around the center that is zeroed out, effectively filtering out those frequency components. The modified frequency domain image G(u,v) is then obtained by element-wise multiplication of F(u,v) with the mask H(u,v):(3)G(u,v)=F(u,v)·H(u,v)

After manipulating the frequency components, the inverse Fourier transform is applied to transform the image back to the spatial domain. The 2D inverse Fourier transform is defined as:(4)$$g\left(x,y\right)=\frac{1}{MN}\sum_{u=0}^{M-1}\sum_{v=0}^{N-1}G\left(u,v\right)e^{2\pi i\left(\frac{ux}{M}+\frac{vy}{N}\right)}$$
where g(x,y) is the resulting image in the spatial domain. Finally, the image is normalized to the range [0,255]. This is achieved by:(5)$$g_{\mathrm{norm}}\left(x,y\right)=\left(\frac{g(x,y)-\min(g)}{\max(g)-\min(g)}\right)\times 255$$
where min(g) and max(g) are the minimum and maximum pixel values in the image g(x,y), respectively. This process is depicted in Figure 3.

This process retains the mean temperature value of an original image while increasing the standard deviation, indicating a greater spread of temperature values and enhanced image features.

Figure 4 demonstrates how ThermoFreq enhances thermal images by manipulating specific frequencies in the MF dataset [49], by plotting the mean and standard deviation of each image in the dataset, before and after applying ThermoFreq. The effect observed in Figure 4 is with a cut-off distance of $D_{0}=1$.

Figure 5 provides a qualitative depiction of the effect of modifying the cut-off distance $D_{0}$ on thermal images sourced from two distinct datasets: MF (left) and PST (right). The images illustrate how the alteration of specific frequencies by ThermoFreq enhances features and contrast within the thermal images. Through adjustment of $D_{0}$, the figure reveals visual alterations in thermal images, underscoring the ability of ThermoFreq to enhance fine details while preserving the overall mean temperature (see also Figure 4).

The observed color variations in the pseudo-color maps, such as increased redness or yellowness in the MF data set, reflect temperature deviations relative to ambient conditions, where the shades of pseudo-color correlate with the thermal contrast of the object against its surroundings. In contrast, the PST dataset, comprising underwater thermal scenes, exhibits minimal visual changes due to inherently constrained temperature gradients in aquatic environments. Although all processing was performed on the raw thermal data, we show the pseudo-color maps for visual illustration and further contextualize these observations; the processed images demonstrate ThermoFreq’s adaptive enhancement capabilities sufficiently. This aligns with the technique’s design to amplify discriminative features while mitigating noise, even when ambient thermal profiles limit observable variability.

Figure 6 represents temperature data in the 3*^rd^* dimension to elucidate the effects of varying cut-off distances $D_{0}$ on discrete thermal images sourced from two datasets: MF (upper) and PST (lower). These images demonstrate the manner in which ThermoFreq adjusts specific frequencies to enhance distinct features and contrast. Through modulation of $D_{0}$, the qualitative influence on thermal images is reflected, underscoring the technique’s ability to augment features and enhance the overall saliency of the images.

## 5. Experiments

We first evaluate the effectiveness of ThermoFreq using existing RGBT segmentation models. Next, we evaluate the impact of the tolerance threshold *t* on segmentation performance across different datasets. Lastly, we evaluate the impact of the tolerance threshold *t* on individual classes of the data set RGBT-Textile. For evaluation, we used two transformer-based RGBT models, viz. CRM_RGBTSeg [34] and MMSFormer [11], which have demonstrated state-of-the-art performance for the RGBT segmentation task. These models are variations of the Swin Transformer models [50] adapted to handle an additional channel of thermal images. In addition, we also report the performance using U-Net [51] models with ResNet18 and ResNet50 [52] backbones. We adopt three commonly used metrics for evaluation: mean Intersection over Union (mIoU), mean precision, and mean recall [18,20,20]. Our model is implemented using PyTorch v1.9.0 framework [53], PyTorch lighting [54], and MMCV libraries [55], and trained on an NVIDIA GeForce RTX 3090 GPU. We follow the training and testing settings from [12,34] and resize the training and testing images to 224×224.

## 6. Results and Discussion

We first evaluate the effectiveness of ThermoFreq and the impact of the tolerance threshold *t* on the performance of semantic segmentation models, specifically CRM_RGBTSeg, MMSFormer, and CNN-based models (UNet with ResNet18 and ResNet50 backbones), across RGBT datasets. For this study, we limit our models to CNN [52,56,57,58] and transformer-based architecture [11,59] due to their high generalization capability. The results in Table 2 and Table 3 show performance gains when models are trained using ThermoFreq, with transformer-based models generally outperforming CNN-based models by 1–2%.

### 6.1. Evaluation of ThermoFreq with Transformer and CNN-Based Models

Table 2 shows the Mean Intersection over Union (mIoU) for CRM_RGBTSeg using Swin Base, Swin Small, Swin Tiny, and UNet with ResNet18 and ResNet50 backbones, both with and without normalization. Transformer-based models consistently outperform CNN-based models across different datasets by approximately 1–2%. For example, on the KAIST Multispectral Pedestrian Dataset, CRM_RGBTSeg with Swin Base achieves an mIoU of 0.567 with normalization, compared to 0.555 with ResNet50 and 0.553 with ResNet18. This suggests that transformer-based models are more robust in learning complex features from RGBT data.

ThermoFreq normalization improves mIoU for all versions of CRM_RGBTSeg and CNN-based models. In particular, Swin Tiny sees a significant improvement, with mIoU increasing from 0.512 to 0.554. In contrast, ResNet-based models show more modest gains, such as an increase from 0.542 to 0.550 for ResNet18. The improved performance of transformer models is attributed to their better ability to model global dependencies and integrate information across multiple modalities, such as RGB and thermal images, while CNN-based models are more reliant on local features.

The performance of MMSFormer with and without normalization is also presented in Table 2. The results show that MMSFormer, although improved with normalization, still lags behind CRM_RGBTSeg by a small margin. For example, on the KAIST Multispectral Pedestrian Dataset, MMSFormer achieves an mIoU of 0.561 with normalization, while UNet with ResNet50 achieves 0.552, highlighting the superior performance of transformer-based architectures even when compared to CNN models. The results indicate that the complementary random masking strategy in CRM_RGBTSeg is more effective for RGBT segmentation than the residue-based approach in MMSFormer.

### 6.2. Impact of Different Tolerance Thresholds

Table 3 shows the mIoU for CRM_RGBTSeg and MMSFormer with different tolerance thresholds *t* (0, 0.5, 1, 5, 10). We observe that *t* values affects model performance across datasets. For example, the KAIST multispectral pedestrian data set benefits from higher *t* values, achieving the highest mIoU of 0.567 at t=10 for CRM_RGBTSeg. Conversely, the PST900 dataset, which has a lower temperature range, achieves its best mIoU of 0.885 at a lower *t* value of 0.5 for CRM_RGBTSeg. This indicates that datasets with a higher thermal temperature range benefit more from higher *t* values, as more aggressive denoising is necessary to handle the increased noise. On the other hand, datasets with a lower temperature range require lower *t* values to avoid losing important information during denoising.

The findings suggest that a small amount of normalization (tolerance level of 0.5) has little effect, while a moderate level (tolerance level of 1) generally improves performance by removing noise. However, high tolerance levels (5 or 10) can be detrimental, especially for datasets with small temperature ranges, as they discard critical information. This effect is more pronounced for smaller models, whereas larger models inherently ignore noise during training. In summary, moderate normalization enhances segmentation performance, particularly for smaller models and datasets with large temperature ranges.

### 6.3. Class Specific Performance on RGBT-Textile Dataset

Table 4 highlights the segmentation performance of CRM_RGBTSeg, MMSFormer, and UNet-ResNet18/ResNet50 on the RGBT-Textile dataset with background, garment, and damaged area as semantic classes, across different thresholds *t* (t=0,0.5,1,5,10). The results provide insights into the characteristics and challenges related to the detection of garment damage. The mIoU values for both models indicate that the dataset is moderately challenging, with the highest performance metrics peaking at t=0.5 and t=1. This suggests that complexity of the dataset allows models to achieve good segmentation accuracy with relatively short processing times, but prolonged processing does not necessarily enhance performance and may introduce noise or overfitting. The precision and recall metrics follow a similar trend as mIoU, with optimal values at t=0.5 and 1, and a gradual decline at t=5 and t=10.

The mIoU values for CRM_RGBTSeg, MMSFormer, UNet-ResNet50, and UNet-ResNet18 show that the dataset is moderately challenging, with the best results generally achieved at t=0.5 and t=1. UNet-ResNet50 shows competitive performance, especially for the background class, while CRM_RGBTSeg performs best overall for the damage area class. Precision and recall metrics follow similar trends, with peak performance at t=0.5 and t=1, and a decline for t=5 and t=10, suggesting that higher tolerance levels can introduce noise.

#### 6.3.1. Background

The background class shows the highest segmentation performance across all models, with CRM_RGBTSeg achieving an mIoU between 0.850 and 0.870 and MMSFormer between 0.836 and 0.854. UNet-ResNet50 outperforms the other models with an mIoU of up to 0.883 at t=1, indicating that it can capture the relatively distinct background features of the dataset well.

#### 6.3.2. Garment

The performance for the garment class is slightly lower than for the background, with CRM_RGBTSeg achieving an mIoU up to 0.830, and UNet-ResNet50 performing best with an mIoU of 0.849. This reflects the more variable nature of garments, including different textures and colors, increasing the difficulty of segmentation.

#### 6.3.3. Damaged Area

The most challenging class is the damaged area, with mIoU values reaching 0.645 for CRM_RGBTSeg and 0.675 for UNet-ResNet50 at t=1. The lower performance suggests that the damaged areas are harder to distinguish due to their irregular shapes and smaller presence in the dataset.

The varying mIoU values between classes indicate that the RGBT-Textile data set contains a range of variability and complexity. Although the background and garment are segmented more reliably by the models, damaged areas present more challenges. This variability necessitates robust segmentation models that can generalize well across different classes and advance the state of damage detection. The lower performance for the damage area class highlights the need to further refine the normalization technique that can specifically enhance the corresponding features. Enhancing the quality and quantity of damaged area annotations, augmenting the dataset with more diverse examples of damaged areas, could further help improve segmentation performance.

These findings indicate that the UNet-ResNet50 model offers competitive performance across all semantic classes, particularly excelling in the background and garment segmentation tasks. The CRM_RGBTSeg model remains strong for damage area detection. Both models benefit from normalization at moderate tolerance thresholds (t=0.5 and t=1). Further improvements could be achieved by increasing the diversity of garment and damage area samples in the dataset and enhancing the models’ ability to capture subtle features in challenging classes like the damaged area.

In the context of textile semantic segmentation tasks, among Swin-based Transformers such as CRM_RGBTseg, CNN-based models, and other Transformer models such as MMSFormer and UNet with ResNet backbones, each offers distinct advantages and challenges. Transformer models excel at modeling long-range dependencies and capturing global context, which translates into superior performance for tasks involving complex data modalities, such as RGB-Thermal (RGBT) segmentation. They consistently outperform CNNs by 1–2% in mean Intersection over Union (mIoU) across various datasets, as shown in Table 4. However, this performance comes at the cost of increased training time and a significantly higher number of parameters, making Transformer models computationally more expensive and difficult to deploy in resource-constrained environments. On the other hand, CNN-based models such as UNet with ResNet18 and ResNet50 are more efficient in terms of training speed and memory usage, making them easier to deploy and maintain in production systems. Despite their efficiency, CNNs are more reliant on local feature extraction, which can limit their ability to capture global context as effectively as Transformers. In summary, Transformer models offer better performance for complex segmentation tasks, but at the cost of higher computational overhead, while CNNs provide a more lightweight and deployable option with slightly lower accuracy.

### 6.4. Robustness to Synthetic Thermal Noise

To ascertain the efficacy of ThermoFreq under various ambient conditions, we enhanced all datasets with thermal-specific perturbations derived from prior work on thermal robustness [60]. These modifications consisted of: (i) Thermal Noise: The sensitivity of a thermal infrared imaging camera is determined by a specification known as the noise equivalent temperature difference (NETD), which specifies the minimum discernible temperature difference that the camera can detect with reliability. To simulate this form of noise, a random temperature value (κ) was introduced to each pixel such that $0<\kappa <Th_{NETD}^{max}$, with the maximum NETD value, $Th_{NETD}^{max}$, established as 0.1 °C, reflecting the worst-case specification criteria for a low-cost thermal imaging camera [60]. (ii) Random Thermal Occlusion: Masking 10–20% of regions with synthesized patches exhibiting extreme low or high temperatures. Subsequently, the models were retrained using these augmented data, while maintaining the integrity of the test sets.

Table 5 shows that ThermoFreq-enhanced models maintain superior performance under synthetic noise compared to baselines. For example, CRM_RGBTSeg + ThermoFreq achieves 0.843 mIoU on augmented RGBT-Textile vs. 0.821 for vanilla CRM_RGBTSeg, an improvement of 2.2%. CNN-based models exhibit larger performance drops (up to 4.8% for UNet-ResNet18), indicating greater sensitivity to thermal perturbations without explicit normalization.

This demonstrates ThermoFreq’s practical value for real-world deployment where ambient-temperature variations induce thermal noise. By explicitly normalizing thermal responses during preprocessing, our method reduces the dependency on data augmentation for robustness.

### 6.5. Comparative Analysis

Existing RGBT models, such as CRM_RGBTSeg and MMSFormer, demonstrate reduced efficacy on RGBT-Textile compared to pedestrian-centric datasets (e.g., KAIST, LLVIP). For example, irregular damage geometries (e.g., jagged tears) yield 12–18% lower precision in transformer-based models due to their dependence on global context, which struggles with localized anomalies [21]. Similarly, CNN architectures such as UNet-ResNet50 exhibit limited sensitivity to subtle thermal gradients (δT < 1.0 °C), critical for detecting stains or discolorations in synthetic fabrics [15]. These limitations highlight the need for domain-specific adaptations, such as ThermoFreq’s frequency-sensitive normalization, to address textile-specific noise and variability.

## 7. Conclusions

In this study, we present the dataset RGBT-Textile, crafted specifically for the semantic segmentation of textile damages. Through a comprehensive series of experiments, we assessed the performance of this dataset by employing benchmark methodologies such as the CRM_RGBTSeg and MMSFormer models. These models were evaluated with and without the implementation of our novel frequency-based normalization technique, ThermoFreq. The empirical findings illustrate that the ThermoFreq normalization technique considerably enhances the performance metrics of the RGBT models. Specifically, we noticed performance improvements ranging from 1% to 1.5% mIoU in different datasets. This improvement trend is consistent, albeit slightly diminished (approximately 0.5 to 0.7) for models trained in the absence of ThermoFreq. The ThermoFreq technique effectively mitigates the inherent noise present in thermal images, thus ensuring robust feature extraction and facilitating effective data fusion. This results in a better and more efficient fusion of RGB and thermal data, culminating in superior model performance in challenging segmentation scenarios. In summary, our RGBT-Textile dataset, in conjunction with the ThermoFreq normalization technique, establishes a strong foundation for advancements in the field of textile damage detection. These contributions underscore the potential for the integration of RGB and thermal data to improve segmentation accuracy, paving the way for more sustainable and automated textile repair solutions. Future research efforts will aim to expand the dataset with additional samples and labels, further investigate the advantages of our normalization technique, and promote collaboration within the computer vision and textile repair communities.

## Figures and Tables

**Figure 1 sensors-25-02306-f001:**
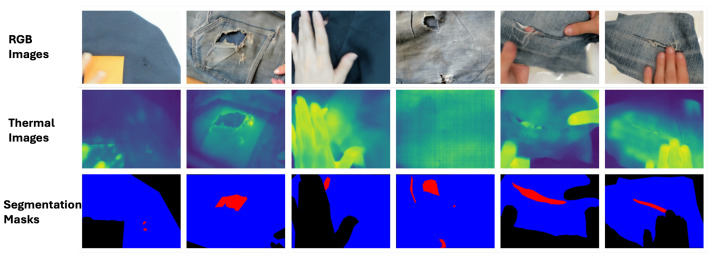
Sample RGB and thermal images with the corresponding segmentation masks from the RGBT-Textile.

**Figure 2 sensors-25-02306-f002:**
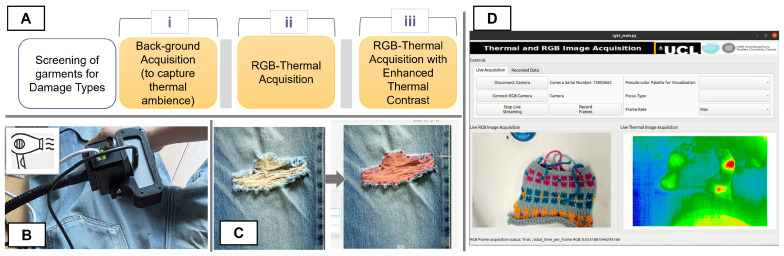
Data acquisition process. (**A**) Acquisition protocol; (**B**) RGBT camera setup; (**C**) Damaged garment and corresponding labeling with segmentation mask; and (**D**) Python-QT based application interface for synchronized RGBT acquisition.

**Figure 3 sensors-25-02306-f003:**
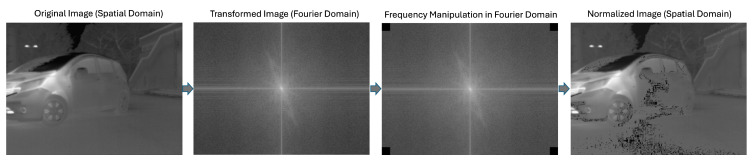
Overview of ThermoFreq Normalization Process.

**Figure 4 sensors-25-02306-f004:**
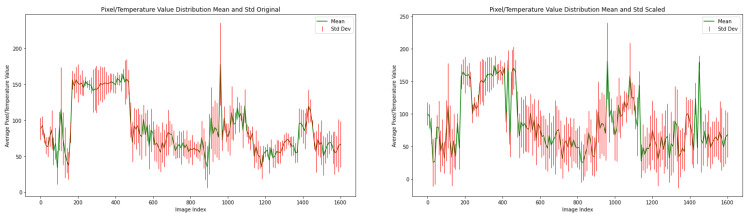
Mean and Standard deviation of MF dataset. Plot on the left shows the mean and std before applying ThermoFreq, and the right shows the mean and std after applying ThermoFreq.

**Figure 5 sensors-25-02306-f005:**
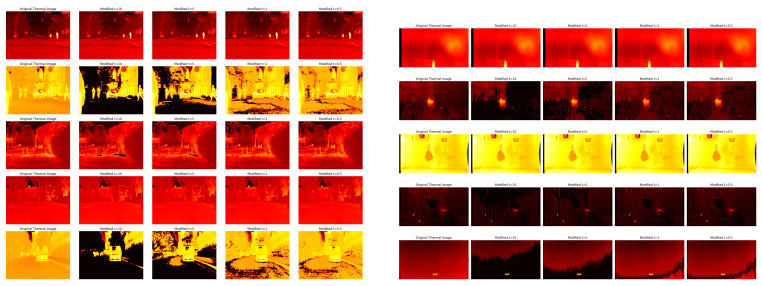
Thermal images from the MF (**left**) and PST (**right**) datasets, illustrating the visual effects of different cutoff distances $D_{0}$.

**Figure 6 sensors-25-02306-f006:**
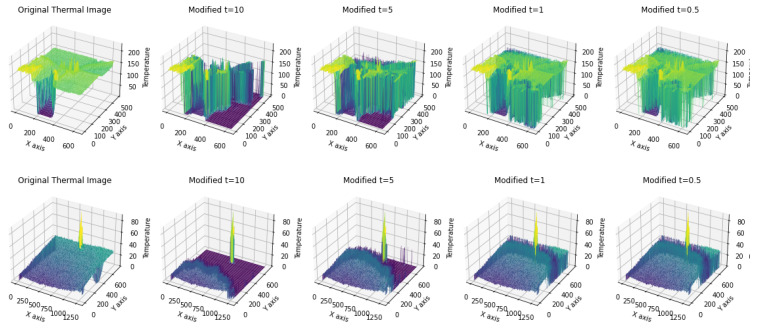
Three dimensional plots for thermal images from the MF (**top**) and PST (**bottom**) datasets, to visualize the differences in the distribution of temperature values across the spatial plane, as a result of applying different cutoff distances $D_{0}$. The brighter yellow shades indicate high temperature values, while the darker blue shades indicate low temperature values.

**Table 1 sensors-25-02306-t001:** Damage characteristics in the RGBT-Textile dataset, including size ranges, material types, thermal variations, and occurrence frequencies. Thermal response variability (δT = 0.5–3.0 °C) complicates segmentation due to overlapping temperature profiles [16].

Damage Type	Size Range (mm^2^)	Material Type	Thermal Variation (°C)	Occurrence Frequency (%)
Holes	5–50	Cotton, Blended	1.2–2.5	30
Stains	10–120	Polyester, Synthetic	0.5–1.8	25
Tears	15–150	All materials	1.0–3.0	20
Stitching Errors	5–30	Cotton, Synthetic	0.7–1.5	15
Discolorations	20–100	Blended, Polyester	0.6–1.2	10

**Table 2 sensors-25-02306-t002:** Comparison of mean Intersection over Union (mIoU) for RGBT datasets using CRM_RGBTSeg and MMSFormer, both with and without normalization. Results highlight that normalization consistently improves the performance. The performance of CNN-based UNet models are additionally shown to compare against transformer-based models.

Model	CRM_RGBTSeg						MMS- Former-84		UNet	
Backbone	Swin Base		Swin Small		Swin Tiny				ResNet18	ResNet50
Normalization		✓		✓		✓		✓	✓	✓
**Datasets**										
KAIST Multispectral Pedestrian	0.552	**0.567**	0.544	0.560	0.512	0.554	0.552	0.565	0.550	0.555
LLVIP Dataset	0.682	**0.701**	0.680	0.692	0.664	0.679	0.663	0.675	0.670	0.673
MF Dataset	0.578	0.586	0.572	0.582	0.556	0.570	0.578	**0.589**	0.577	0.580
OSU Thermal Pedestrian	0.980	**0.994**	0.980	**0.994**	0.970	**0.994**	0.980	0.992	0.990	0.993
McubeS	0.523	**0.545**	0.519	0.542	0.497	0.530	0.504	0.522	0.512	0.520
PST900	0.877	**0.889**	0.871	0.885	0.849	0.880	0.874	0.855	0.872	0.875
RGBT-Textile (Ours)	0.842	0.846	0.840	0.843	0.822	0.835	0.842	**0.852**	0.840	0.845

**Table 3 sensors-25-02306-t003:** Comparison of mean Intersection over Union (mIoU) for RGBT datasets using CRM_RGBTSeg, MMSFormer, and UNet-ResNet18/ResNet50, with different *t* thresholds (*t* = 0, 0.5, 1, 5, 10). The results highlight the performance improvements with normalization for both models across different datasets, indicating their robustness and accuracy in RGBT semantic segmentation tasks.

Dataset\Model	CRM_RGBTSeg					MMSFormer					Unet ResNet18					Unet ResNet50				
**Tolerance**	t=0	t=0.5	t=1	t=5	t=10	t=0	t=0.5	t=1	t=5	t=10	t=0	t=0.5	t=1	t=5	t=10	t=0	t=0.5	t=1	t=5	t=10
KAIST Multispectral Pedestrian	0.552	0.563	**0.564**	0.463	0.367	0.552	0.561	**0.562**	0.464	0.365	0.550	0.560	**0.561**	0.460	0.365	0.555	0.565	**0.566**	0.462	0.368
LLVIP Dataset	0.682	0.697	**0.698**	0.600	0.501	0.663	0.671	**0.672**	0.574	0.475	0.670	0.684	**0.685**	0.589	0.482	0.673	0.688	**0.689**	0.590	0.485
MFNet Dataset	0.556	0.578	**0.579**	0.480	0.381	0.578	0.585	**0.586**	0.487	0.389	0.577	0.595	**0.596**	0.489	0.387	0.580	0.598	**0.599**	0.491	0.390
OSU Thermal Pedestrian	0.980	0.990	**0.991**	0.793	0.694	0.980	0.988	**0.989**	0.791	0.692	0.990	**0.993**	0.994	0.790	0.693	0.993	**0.996**	0.996	0.792	0.695
McubeS	0.523	0.541	**0.542**	0.444	0.345	0.504	0.518	**0.519**	0.421	0.322	0.512	0.529	**0.531**	0.422	0.327	0.520	0.535	**0.537**	0.425	0.330
PST900	0.877	**0.885**	0.868	0.748	0.639	0.874	**0.881**	0.872	0.729	0.625	0.872	**0.883**	0.870	0.735	0.628	0.875	**0.886**	0.873	0.737	0.630
Dataset (Ours)	0.822	0.842	**0.843**	0.745	0.646	0.842	0.848	**0.849**	0.751	0.652	0.840	0.856	**0.857**	0.749	0.650	0.845	0.861	**0.862**	0.754	0.655

**Table 4 sensors-25-02306-t004:** Performance comparison of CRM_RGBTSeg, MMSFormer and U-Net models on RGBT-Textile dataset for different *t* thresholds (t=0,0.5,1,5,10). Metrics include mean Intersection over Union (mIoU), precision, and recall for the entire dataset and individual classes (background, garment, damaged area).

Performance Metrics\Model	CRM_RGBTSeg					MMSFormer					UNet-ResNet18					UNet-ResNet50				
**Tolerance**	t=0	t=0.5	t=1	t=5	t=10	t=0	t=0.5	t=1	t=5	t=10	t=0	t=0.5	t=1	t=5	t=10	t=0	t=0.5	t=1	t=5	t=10
mIoU/Whole dataset	0.822	**0.842**	0.841	0.830	0.820	0.810	**0.828**	0.827	0.817	0.808	0.840	0.856	**0.857**	0.749	0.650	0.845	0.861	**0.862**	0.754	0.655
Precision/Whole dataset	0.830	**0.850**	0.849	0.838	0.828	0.818	**0.835**	0.834	0.824	0.815	0.848	0.864	**0.865**	0.757	0.656	0.853	0.870	**0.871**	0.759	0.661
Recall/Whole dataset	0.815	**0.835**	0.834	0.823	0.815	0.803	**0.820**	0.819	0.811	0.803	0.832	0.849	**0.850**	0.738	0.646	0.837	0.854	**0.855**	0.742	0.650
mIoU/Background	0.850	**0.870**	0.869	0.858	0.850	0.836	**0.854**	0.853	0.845	0.838	0.860	0.878	**0.879**	0.769	0.661	0.865	0.882	**0.883**	0.773	0.665
mIoU/Garment	0.810	**0.830**	0.829	0.820	0.810	0.796	**0.813**	0.812	0.808	0.798	0.825	0.842	**0.843**	0.740	0.633	0.830	0.848	**0.849**	0.745	0.640
mIoU/Damaged area	0.580	**0.645**	0.640	0.630	0.620	0.565	**0.629**	0.623	0.617	0.608	0.590	0.655	**0.660**	0.550	0.448	0.605	0.670	**0.675**	0.555	0.452

**Table 5 sensors-25-02306-t005:** Performance comparison (mIoU) on original vs. thermally-augmented datasets. Models trained with ThermoFreq show greater robustness to synthetic noise/occlusions compared to baselines.

Dataset	CRM_RGBTSeg		MMSFormer		UNet-ResNet50	
	Orig	Aug	Orig	Aug	Orig	Aug
KAIST Multispectral Pedestrian	0.567	0.541	0.565	0.538	0.555	0.512
LLVIP Dataset	0.701	0.683	0.675	0.651	0.673	0.634
MF Dataset	0.586	0.569	0.589	0.563	0.580	0.547
RGBT-Textile (Ours)	0.846	0.843	0.852	0.831	0.845	0.809

## Data Availability

The dataset is available at this download link https://drive.google.com/drive/folders/1HLri3SDPHSY0AsAmaEloh_0Q13E91u2v (accessed on 28 March 2025).

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
