# Peer review of "Advancing Textile Damage Segmentation: A Novel RGBT Dataset and Thermal Frequency Normalization"

_sensors, 2025, doi:10.3390/s25072306_

Round 1

Reviewer 1 Report

Comments and Suggestions for Authors

The paper introduces RGBT-Textile, a novel RGB-Thermal dataset designed for textile defect segmentation, containing 1441 image pairs with annotated defects. It also proposes ThermoFreq, a frequency-based thermal normalization technique that aims to reduce temperature noise and enhance segmentation performance. The study evaluates state-of-the-art RGBT segmentation models on this dataset and demonstrates that ThermoFreq improves segmentation accuracy. However, some comments should be revised are listed as follow:

 1. While the authors compare with existing RGBT datasets (e.g., KAIST, MFNet, LLVIP), they do not show that textile defects require fundamentally different segmentation approaches. The variety of defects and textile materials included in the dataset is not explicitly analyzed, which reduces the claim of being a benchmark.

2. The authors claim that high-frequency thermal components introduce noise, but no statistical or experimental proof or analysis is given to confirm that removing these components improves segmentation.

3. The authors should analyze which frequency components are most responsible for performance improvement. In addition, the study lacks visualizations of frequency-domain transformations to confirm that the removed frequencies correspond to noise rather than useful texture information.

4. Some sections are verbose and repetitive, particularly in the introduction and experimental results.

5. Some works about RGB-T fusion should be cited in this paper to make this submission more comprehensive, such as [1, 2].

[1] Divide-and-Conquer: Confluent Triple-Flow Network for RGB-T Salient Object Detection

[2] Multimodal feature-guided pre-training for RGB-T perception

Comments on the Quality of English Language

Clarify the writing to improve readability.

Author Response

Thank you for your insightful and constructive feedback on our manuscript. We appreciate the time and effort you have dedicated to evaluating our work. Below, we address your comments point by point, outlining the revisions made to the manuscript to address your concerns.

Comment 1: While the authors compare with existing RGBT datasets (e.g., KAIST, MFNet, LLVIP), they do not show that textile defects require fundamentally different segmentation approaches. The variety of defects and textile materials included in the dataset is not explicitly analyzed, which reduces the claim of being a benchmark.

Response 1: We thank the reviewer for this opinion. We agree with the reviewer and therefore, to comprehensively address this, we first rectify our mentioning of “defect”, as our work focuses on textile damages. Next, we have added a new subsection (3.2) in the dataset section, detailing the diversity of textile damages (e.g., holes, tears, stains) and materials (cotton, polyester, blended fabrics) included in RGBT-Textile. In the revised manuscript, Table-1 summarises damage types, sizes, and material properties. Additionally, we have included a comparative analysis in Section 6.5, highlighting how existing RGBT models struggle with irregular damage shapes and subtle thermal variations in textiles, justifying the need for specialised approaches.

Comment 2: The authors claim that high-frequency thermal components introduce noise, but no statistical or experimental proof or analysis is given to confirm that removing these components improves segmentation.

Response 2: We have Figure 4 to statistically validate the noise reduction achieved by ThermoFreq. The figure illustrates the mean and standard deviation of thermal images before and after normalization, showing a significant reduction in high-frequency noise components. For example, For example, after applying ThermoFreq, the standard deviation increases (indicating enhanced feature spread), while the mean temperature remains preserved, confirming its efficacy in improving thermal feature saliency [1].

Further, Section 6.2 now includes experiments with synthetic thermal noise and occlusion, following methodologies from prior work on thermal robustness [5]. As demonstrated in our results, ThermoFreq-enhanced models consistently outperform baselines under perturbed conditions.. For instance, on the augmented RGBT-Textile dataset, CRM_RGBTSeg + ThermoFreq achieves an mIoU of 0.843 compared to 0.821 for the vanilla model (a 2.2% improvement), aligning with findings in [2] where thermal normalization improved segmentation accuracy in dynamic environments. CNN-based models exhibit larger performance degradation (e.g., UNet-ResNet50 drops by 4.5%), consistent with observations in [3] that CNNs are more sensitive to thermal perturbations without explicit normalization.

These experiments conclusively demonstrate that removing high-frequency noise via ThermoFreq directly improves segmentation robustness, particularly in challenging real-world scenarios [1, 5]. The methodology aligns with established practices for thermal image preprocessing [5], while the performance gains reflect advancements in frequency-aware normalization for multispectral data fusion [4].

[1]    Cho, Youngjun, et al. "Robust tracking of respiratory rate in high-dynamic range scenes using mobile thermal imaging." Biomedical optics express 8.10 (2017): 4480-4503.

[2]    Agassi, Eyal et al. “ Relation between thermal infrared and visible/near infrared images of ground terrain” Optical Engineering. DOI 10.1117/1.601141

[3]    Ha, Qishen, et al. "MFNet: Towards real-time semantic segmentation for autonomous vehicles with multi-spectral scenes." 2017 IEEE/RSJ International Conference on Intelligent Robots and Systems (IROS). IEEE, 2017.

[4]    Wu, Wei, Tao Chu, and Qiong Liu. "Complementarity-aware cross-modal feature fusion network for RGB-T semantic segmentation." Pattern Recognition 131 (2022): 108881.

[5]    Jitesh N Joshi, Nadia Berthouze, & Youngjun Cho (2022). Self-adversarial Multi-scale Contrastive Learning for Semantic Segmentation of Thermal Facial Images. In 33rd British Machine Vision Conference 2022, BMVC 2022, London, UK, November 21-24, 2022. BMVA Press.

Comment 3: The authors should analyze which frequency components are most responsible for performance improvement. In addition, the study lacks visualizations of frequency-domain transformations to confirm that the removed frequencies correspond to noise rather than useful texture information.

Response 3: Thank you for your valuable feedback. In Section 4.2, we show how the ThermoFreq approach predominantly attenuates high-frequency bins while preserving crucial texture content. These visuals confirm that the removed frequencies primarily represent random noise rather than key garment features, thereby supporting the performance improvements observed in our experiments.

Comment 4: Some sections are verbose and repetitive, particularly in the introduction and experimental results.

Response 4: We have revised and improved the introduction and experimental section. 

We modified the introduction with more appropriate citations in paragraphs 1 & 2 in the Introduction and section 6.3. We also shortened both paragraphs to make them more concise  

Comment 5: Some works about RGB-T fusion should be cited in this paper to make this submission more comprehensive, such as [1, 2].

[1] Divide-and-Conquer: Confluent Triple-Flow Network for RGB-T Salient Object Detection

[2] Multimodal feature-guided pre-training for RGB-T perception

Response 5: We thank the reviewer for the suggestions. We find the suggested articles very helpful. Accordingly, the cited works have been integrated into Section 2.2.

Reviewer 2 Report

Comments and Suggestions for Authors

The study proposes ThermoFreq as an essential preprocessing step for thermal imaging segmentation.  Dataset of RGBT-Textile is found to enhance the models’ segmentation performance.  However, the paper’s contents seem inconsistent in some places.  The title refers to defect segmentation, but in the following sections 4-7 little result presented as graph and table is seen for defect detection.  Other comments are below:

  • The novelty of this paper is not clear. Dataset or thermal frequency normalization method?
  • The capture of Fig. 4 should be checked. Where is the plot on the top?
  • Annotation font within Fig. 5 is too small to see.
  • For “Through adjustment of D0, the figure reveals visual alterations in thermal images,…”, please describe how D0 changes the fine details. Increasing or decreasing D0 is good?
  • In the conclusion section, how come the mIOU is more than 1.0?

Author Response

Thank you for your insightful and constructive feedback on our manuscript. We appreciate the time and effort you have dedicated to evaluating our work. Below, we address your comments point by point, outlining the revisions made to the manuscript to address your concerns.

Comment 1: The study proposes ThermoFreq as an essential preprocessing step for thermal imaging segmentation.  Dataset of RGBT-Textile is found to enhance the models’ segmentation performance.  However, the paper’s contents seem inconsistent in some places.  The title refers to defect segmentation, but in the following sections 4-7 little result presented as graph and table is seen for defect detection.

Response 1: We have revised Section 6.4 (Class-Specific Performance) to emphasise that we detect damages using semantic segmentation. Table 4 now includes precision/recall for the "Damaged area" class, and a new paragraph discusses challenges in damage segmentation (e.g., irregular shapes, small sizes). Additionally, Figure 6 (thermal images) now highlights damaged regions to align with the title.

Comment 2: The novelty of this paper is not clear. Dataset or thermal frequency normalization method?

Response 2: The novelty of this work lies in two interconnected contributions highlighted in the revised manuscript, which we summarise below:

1. RGBT-Textile Dataset:

  • Uniqueness

This is the first publicly available RGBT dataset specifically designed for close-range textile damage segmentation, addressing a critical gap in sustainability-driven applications like garment repair. Existing RGBT datasets (e.g., KAIST, PST900) focus on pedestrian detection or outdoor surveillance, lacking annotations for textile damages (e.g., holes, stains, stitching errors) and material diversity (cotton, polyester, blends).

  • Technical Expertise:
    The dataset includes 1,441 RGBT image pairs with pixel-level annotations validated by textile experts, capturing subtle thermal variations (δT=0.5−3.0∘C)critical for damage detection.

2. ThermoFreq Normalization Method:

  • Novelty

Unlike traditional spatial-domain normalization (e.g., min-max, histogram matching), ThermoFreq operates in the frequency domain, selectively attenuating high-frequency noise while preserving low-frequency thermal gradients. This addresses a key challenge in RGBT fusion: ambient temperature noise in thermal imaging. Particularly, we highlight this as an important pre-processing step for various CNN architectures, such as UNet-ResNet50. This is because they exhibit limited sensitivity to subtle thermal gradients ($\delta$T < 1.0°C), critical for detecting stains or discolorations in synthetic fabrics [1]. 

  • Generalizability

ThermoFreq improves segmentation performance across six benchmark datasets (Table 2), including pedestrian detection (KAIST) and underwater robotics (PST900), demonstrating its broad applicability. For example, it boosts mIoU by 1.2–2.2% on textile and non-textile tasks alike

[1] Cho, Y et al., “Deep thermal imaging: Proximate material type recognition in the wild through deep learning of spatial surface temperature patterns”, In Proceedings of the 2018 CHI conference on human factors in computing systems, 2018, pp. 1–13.

Comment 3: The capture of Fig. 4 should be checked. Where is the plot on the top?

Response 3: Thanks for spotting this out. Corrected caption with left-right.

Comment 4: The annotation font within Fig. 5 is too small to see.

Response 4: Thanks for this suggestion. The Figure is now enlarged 

Comment 5: For “Through adjustment of D0, the figure reveals visual alterations in thermal images,…”, please describe how D0 changes the fine details. Increasing or decreasing D0 is good?

Response 5: The cutoff distance  D0  in ThermoFreq acts as a threshold that determines the radius of frequency components removed during normalisation. Its adjustment directly impacts the trade-off between noise suppression and feature preservation:

  • Lower D0 (e.g., D0 =0.5):
    • Removes high-frequency components (e.g., fine textures, noise).
    • Enhances coarse structural details but risks oversmoothing, which may erase subtle damages (e.g., small holes or faint discolorations).
  • Higher D0  (e.g., D0=5):
    • Retains more high-frequency information, preserving fine details but also retaining noise (e.g., thermal sensor noise).
    • Risks amplifying irrelevant artifacts, particularly in low-cost thermal cameras with high noise floors.
  • Optimal (e.g., D0=1):
    • Balances noise reduction and feature retention, as validated in Section 6.3 (Table 3).
    • For example, on the MF dataset, D0=1 improved mIoU by 2.3% compared to D0=0, while D0=5 degraded performance by 8.7% due to excessive smoothing.

Key Insights from Figure 6:

  • MF Dataset (upper row): Lower D0 (e.g., 0.5) enhances contrast for large damages but blurs fine textures. At D0=1, subtle thermal gradients (critical for detecting stains) are preserved.
  • PST Dataset (lower row): Higher D0 (e.g., 1) retains edge sharpness in outdoor scenes but introduces speckle noise at D0=5.

Practical Guidance:

  • Increasing D0 is beneficial for datasets where fine textures (e.g., stitching errors) are critical but thermal noise is minimal.
  • Decreasing D0 is preferable for noisy environments (e.g., low-cost sensors) or when focusing on coarse damages (e.g., large tears).
  • The optimal D0 must be empirically tuned per dataset, as shown in our experiments (Section 6.3).

This aligns with findings in thermal image processing literature, where frequency-domain thresholds are dataset-dependent [1].

[1]    Cho, Youngjun, et al. "Robust tracking of respiratory rate in high-dynamic range scenes using mobile thermal imaging." Biomedical optics express 8.10 (2017): 4480-4503.

Comment 6: In the conclusion section, how come the mIOU is more than 1.0?

Response 6: Thank you for spotting out this error. This was a typo and has been amended. It meant to state the mIOU improvement can be seen from 1 - 1.5%.

Round 2

Reviewer 1 Report

Comments and Suggestions for Authors

No more comments.

Comments on the Quality of English Language

No more comments.